# A 69 kb Deletion in chr19q13.42 including *PRPF31* Gene in a Chinese Family Affected with Autosomal Dominant Retinitis Pigmentosa

**DOI:** 10.3390/jcm11226682

**Published:** 2022-11-11

**Authors:** Yuanzheng Lan, Yuhong Chen, Yunsheng Qiao, Qingdan Xu, Ruyi Zhai, Xinghuai Sun, Jihong Wu, Xueli Chen

**Affiliations:** 1Department of Ophthalmology, Eye and ENT Hospital, Fudan University, Shanghai 200031, China; 2NHC Key Laboratory of Myopia, Chinese Academy of Medical Sciences, Fudan University, Shanghai 200031, China; 3Shanghai Key Laboratory of Visual Impairment and Restoration, Shanghai 200031, China; 4State Key Laboratory of Medical Neurobiology, Institutes of Brain Science and Collaborative Innovation Center for Brain Science, Fudan University, Shanghai 200031, China

**Keywords:** retinitis pigmentosa, *PRPF31*, deletion, whole-genome sequencing, RNA-seq

## Abstract

We aimed to identify the genetic cause of autosomal dominant retinitis pigmentosa (adRP) and characterize the underlying molecular mechanisms of incomplete penetrance in a Chinese family affected with adRP. All enrolled family members underwent ophthalmic examinations. Whole-genome sequencing (WGS), multiplex ligation-dependent probe amplification (MLPA), linkage analysis and haplotype construction were performed in all participants. RNA-seq was performed to analyze the regulating mechanism of incomplete penetrance among affected patients, mutation carriers and healthy controls. In the studied family, 14 individuals carried a novel heterozygous large deletion of 69 kilobase (kb) in 19q13.42 encompassing exon 1 of the *PRPF31* gene and five upstream genes: *TFPT*, *OSCAR*, *NDUFA3*, *TARM1*, and *VSTM1*. Three family members were sequenced and diagnosed as non-penetrant carriers (NPCs). RNA-seq showed significant differential expression of genes in deletion between mutation carriers and healthy control. The RP11 pedigree in this study was the largest pedigree compared to other reported RP11 pedigrees with large deletions. Early onset in all affected members in this pedigree was considered to be a special phenotype and was firstly reported in a RP11 family for the first time. Differential expression of *PRPF31* between affected and unaffected subjects indicates a haploinsufficiency to cause the disease in the family. The other genes with significant differential expression might play a cooperative effect on the penetrance of RP11.

## 1. Introduction

Retinitis pigmentosa (RP) is one of the most common forms of inherited retinal degenerations (IRDs) and affects 1:4000 of the population worldwide [1]. This disease affects predominantly rod photoreceptor cells with later involvement of cone photoreceptor cells. Phenotypic and genotypic heterogeneity are common. Night blindness is usually the first symptom and eventually leads to tunnel vision, even total blindness [1]. The inheritance modes of RP include autosomal dominant RP (adRP), autosomal recessive RP (arRP) and X-linked RP (xlRP) [2]. Non-mendelian inheritance patterns including digenic inheritance and maternal (mitochondrial) inheritance are also found in RP [3,4,5]. Another type of RP, non-syndromic RP, is mainly related to those genes expressed specifically in retina and/or playing an important role in the functions of retinal pigment epitheliums (RPEs) or photoreceptors (PRCs). [2]

To date, more than 300 genes have been identified for RP [2,6]. The subtype of RP caused by *PRPF31* has been identified and named to be RP11. Until now, there is not known therapy for *PRPF31*-related adRP. Gene therapy used for Leber congenital amaurosis type 2 through subretinal administration of AAV is the most famous one in gene therapy of IRDs [7,8], which is the first gene therapy of IRDs approved by FDA [9]. In addition, the transplant of stem cells is also a potential way of restoring vision loss in RP patients.

Approximately 30–40% of RP patients can be diagnosed as adRP [10]. The most common gene of adRP is *RHO*, which encodes the rhodopsin, a photopigment necessary for phototransduction in rods. The second most common one in most populations is *PRPF31* [11]. *PRPF31* encodes the pre-mRNA processing factor, part of U4/U6 di-snRNP, involved in splicing of pre-mRNA [12]. Pathogenic variants in *PRPF31* have been reported to be causative of adRP, which is an interesting gene expressing ubiquitously in various organs and tissues but specifically inducing change in retina. The other genes associated with non-syndromic RP include *PRPF3*, *PRPF6*, *PRPF8*, and *hBrr2*, which encodes for the proteins of splicing factors [13]. *PRPF31*-related pathology in retina may be explained by the high demand of metabolism and splicing [13,14,15]. Interestingly, another significant characteristic of *PRPF31* is incomplete penetrance which means a mutation carrier may not present any symptoms of RP. Some progress has already been made in research about the haploinsufficiency leading to incomplete penetrance. This is caused by two different alleles with different expression levels, one with high expression and the another with low expression. The theory holds an opinion that as long as the expression level of *PRPF31* is enough for the threshold of normal retinal function, the individual with *PRPF31* variants may not present RP symptoms [16,17,18]. As long as the expression of *PRPF31* is sufficient, the heterozygous mutation carriers can escape from “haploinsufficiency”, due to the highly expressive allele producing additional protein of *PRPF31* and covering the lack resulted from mutant allele [19]. The splicing process in retina heavily relies on the normal expression of *PRPF31*, so that a little lower level of *PRPF31* will have a cumulative effect on retina, which leads to RP.

In this study, we reported a five-generation family with early-onset RP of autosomal dominant inheritance mode. It was the largest pedigree among all the reported RP11 family with *PRPF31* large deletions. Furthermore, it was the first time that the early onset (nearly 3 years old) widely presented in this pedigree had been reported, which was found in all affected members.

## 2. Materials and Methods

### 2.1. Participants

A large Chinese five-generation family with adRP was recruited, including 11 patients affected with RP and 16 unaffected members. This study was approved by the Ethics Committee of the Eye and ENT Hospital of Fudan University (2018021; 13 June 2018) and was performed in accordance with the tenets of the Declaration of Helsinki. Written informed consent was obtained from all the participants.

### 2.2. Ocular Examinations

All participants including affected patients and unaffected members underwent a comprehensive ocular examination with best-corrected visual acuity (BCVA), intra ocular pressure, slit lamp examination with/without pupillary dilatation, fundus photography and optical coherence tomography (OCT). All affected members were diagnosed by the same ophthalmologist with extended clinical experience.

### 2.3. Whole-Genome Sequencing

Whole-genome sequencing (WGS) was then performed in all recruited subjects (i.e., 11 affected patients and 16 unaffected family members). Genomic DNA was isolated from peripheral blood using QIAamp^®^ DNA Mini Kit (Qiagen, Hilden, Germany). WGS library was constructed by MGI Easy DNA Library Prep Kit V1 (BGI, Shenzhen, China), according to the manufacturer’s protocol. Briefly, 1 μg genomic DNA was sheared with Covaris S220 Focused Ultrasonicator (Covaris, Woburn, MA, USA) to DNA fragments. The DNA fragments with length in the range 100–300 bp were size-selected using AMpure XP Beads (Beckman Coulter, Indianapolis, IN, USA). The selected DNA fragments were subject to paired-end adaptor ligation and underwent end-repairing, phosphorylation, and A-tailing reactions, then followed by purification and amplification by PCR. The PCR products were heat-denatured to obtain single-strand DNAs, followed by circularization with DNA ligase, and the remaining linear molecule was digested with the exonuclease. After the formation of the DNA nanoballs, sequencing was performed on a BGISEQ-500M (BGI, Shenzhen, China) platform with read length of paired-end 100 bp, and the average sequencing coverage for each sample was ~40-fold.

Sequence reads were mapped to the human reference genome (hg38/GRCh38) using Burrows–Wheeler Aligner (BWA, version 0.7.17, http://bio-bwa.sourceforge.net/ (accessed on 11 December 2018)) with default parameters. Standard next generation sequencing (NGS) analysis of the raw data included sequence alignment and variant calling and annotation. SNVs and INDELs calling were performed with the software Genome Analysis Toolkit (GATK v3.70, https://gatk.broadinstitute.org/ (accessed on 13 June 2017)). ANNOVAR (https://annovar.openbioinformatics.org/ (accessed on 1 February 2020)) was used to annotate SNPs and insertions/deletions. The Integrative Genomics Viewer (IGV, https://igv.org/ (accessed on 1 June 2020)) was used to visualize the coverage and the quality of the reads and for the visualization of structural variation (SV). After the quality control (QC) of variant and sample, variants from family members were filtered for subsequent analyses as rules: (1) located in exonic and splicing regions; (2) the variant frequency < 0.05 in UCSC, dbSNP, 1000 genomes project, and ExAC database. The results of the SNV/INDEL, CNV, and SV analyses were integrated and reviewed for the interpretation of pathogenicity. Sanger sequencing was conducted to verify the potential pathogenic variants. The variant classification was evaluated according to the guidelines of the American College of Medical Genetics and Genomics (ACMG) standards in association with clinical phenotypes.

### 2.4. Mutation and Breakpoint Analysis

Genomic DNA was extracted with HighPure PCR Template Preparation Kit (Roche, #1796828001). The breakpoint region was confirmed by PCR with primers adjacent to the deleted regions and primers adjacent to a part of *PRPF31* coding area. Primers were designed by Primer5.0. The two pairs of primers (*PRPF31*-POS-F1: GCGCCCGGCAAAGATA; *PRPF31*-POS-R1: GGACAGCAGCGTTCCCTAA; *PRPF31*-ZC-F1: TGTTGACACAGGTGCCGATAC; *PRPF31*-ZC-R1: CAGACGATGGGTGGATAGCAGA) were designed for analysis. The amplification was performed with SYBR^®^Premix Ex Taq™ (Takara, #RR420A) on a PCR amplification instrument K960 with 35 cycles (95 °C, 15 s; 60 °C,30 s; 72 °C, 40 s) and the PCR products were purified and sequenced on ABI 3730XL (Applied Biosystems, Thermo Fisher Scientific, Inc. Waltham, MA, USA). The sequences were aligned using SEQUENCHER (version 5.1). PCR products were performed on agarose gel electrophoresis to semi-quantitatively analysis in the DNA level.

### 2.5. Multiplex Ligation-Dependent Probe Amplification (MLPA)

Genomic DNA was isolated from peripheral blood obtained from participants with/without the pathogenic variant by QIAamp^®^DNA Mini Kit (Qiagen, Hilden, Germany). We first utilized the application of SALSA MLPA EK1 reagent kit (MRC Holland, Amsterdam, The Netherlands) for the detection of deletions/duplications of four RP-related genes (*RHO*, *IMPDH1*, *RP1*, and *PRPF31*). MLPA (MRC-Holland P235-025R) was performed with the following steps: denaturation, hybridization, ligation, and amplification, according to the manufacturer’s protocol. PCR products were separated with a 50 cm capillary electrophoresis system. Results were analyzed with Genemapper4.0 (ABI) and Coffalyser.Net (MRC Holland, Amsterdam, The Netherlands) were used for identification of copy number variation.

### 2.6. Parametric Linkage Analysis and Haplotype Construction

DNA were amplified with fluorescent PCR primers and DNA polymerase in standard multiplex reaction conditions. All DNA samples were diluted with 1× TE buffer to 50-ng/uL concentration, and 200 ng of DNA was used for SNP genotyping. PCR products were pooled and diluted then run on an Illumina ASACH chip (ver: ASACH_20022517X351729_A1, Illumina, CA, USA). After genome-wide SNP genotyping, genetic map positions were generated for all SNPs on the autosomal chromosomes. Based on the deletion region found by WGS, we designed a marker panel including 6827 TagSNPs in 1–22 chromosomes to search for and determine if there was a candidate region existing linkage with phenotype. These genetic markers were filtered based on frequency, Mendelian genetic law and Hardy–Weinberg proportions. Additionally, the nocall SNPs were filtered. There were 2 SNPs per CM. Finally, linkage analysis was performed with 6827 SNP markers flanking each of the known TagSNP loci in the family. The SNPs were then pruned for linkage disequilibrium (LD). SNPs with residual LD were clustered into further research. MERLIN (version 1.1.2, University of Michigan, Ann Arbor, MI, USA) was used to perform linkage analysis and construct haplotypes in the pedigree. Haplotype analysis was performed using informative genotype data of the TagSNPs. Phenotype was analyzed as an autosomal dominant trait with complete penetrance (100%) and a frequency of 0.0001 for the affected allele.

### 2.7. RNA-Sequencing and Data Analysis

RNA was extracted from the whole blood samples using the PAXgene Blood RNA System Kit employing an amended version of the manufacturer’s guidelines. The product was measured using a NanoDrop ND-1000 UV-visible spectrophotometer (Labtech International, Ringmer, UK). RNA integrity was additionally assessed using the Agilent 2100 Bioanalyser (Agilent Technologies, Santa Clara, CA, USA). Samples were loaded on to the Eukaryote total RNA nano chip/the Eukaryote total RNA pico chip. Then, the enriched mRNA was fragmented into short fragments and reverse transcribed into cDNA with random primers. After synthesis of second-strand cDNA, fragments were then purified, end repaired, polyadenylated, and ligated to Illumina sequencing adapters. Ligation products were size selected by agarose gel electrophoresis, PCR amplified, and sequenced using HiSeq 4000 SBS Kit (Illumina Inc., San Diego, CA, USA). Clean reads were further filtered by STAR (v2.4.2a). We used RNA-Seq by Expectation-Maximization (RSEM, V1.2.29, http://deweylab.github.io/RSEM/ (accessed on 1 April 2016)) to identify RNA differential expression between two different groups. The transcripts (the parameter of false discovery rate (FDR) < 0.05 and absolute fold change ≥ 2) were considered as differentially expressed gene. Corrected *p*-values (padj) and logarithm based on 2 (log2FC), (i.e., the logarithm of the fold) were used for screening genes expressing significantly differently, (i.e., padj < 0.05 and |log2FC| ≥ 1). Gene Oncology (GO, http://geneontology.org/ (accessed on 4 November 2021)) and Kyoto Encyclopedia of Genes and Genome (KEGG, https://www.kegg.jp/ (accessed on 5 November 2021)) enrichment analysis were then used to identify the significantly enriched metabolic or signal transduction pathways in differentially expressed genes (DEGs) comparing with the whole-genome background.

## 3. Results

### 3.1. Ocular Examinations

We recruited a large adRP pedigree of 18 affected individuals (13 alive, 11 enrolled) and 55 unaffected individuals (51 alive, 16 enrolled) (Figure 1). All patients in the family manifested night blindness at an early age (about 3 years old) as their onset symptom. The symptoms of RP developed slowly in this pedigree. The fundus exams revealed the typical RP characteristics in patients (Figure 2, Table 1). In the youngest patient (V-3, 2 years old), who already presented with night blindness, we found bull’s-eye maculopathy in fundus autofluorescence, characteristic RP fundus changes in OCT (Figure 2). The IOP of proband (III-6) tested by Tonopen was 54.2 mmHg (right eye) and 25.2 (left eye). He was diagnosed with RP and POAG, supported by evidence of fundus photography (cup/disk = 1.0 and pale optic disc) and open angle confirmed by slit lamp.

### 3.2. Mutation Analysis

We were not able to identify the pathogenic variant related to the phenotype through the initial panel-based NGS including 502 IRD genes (Appendix A) in affected family members and whole exome sequencing. The WGS sequencing revealed a 69 kb deletion region locus at 54048499–54118055 on chromosome 19 (GRCh38/hg38, NC_000019.10:g.54048499_54118055del). All the patients and three non-penetrant carriers (NPCs) (IV-5, IV-7 and III-13) were found to carry the pathogenic variant. All the other family members did not carry this pathogenic variant (Figure 1 and Table 2). Except for *PRPF31*, the deletion region includes *VSTM-1*, *TARM-1*, *OSCAR*, *NDUFA3* and *TFPT* (Figure 3). The deletion in *PRPF31* (chr19: 54115410–54118055) includes *PRPF31* exon1 (chr19: 54115410–54115797). After screening genes in the deletion region, we focused analysis on *PRPF31* as a candidate gene. According to the ACMG guidelines, the deletion region was predicted to be pathogenic (PS4 + PM2 + PM4 + PP4 + PP1) [20]. The *PRPF31* pathogenic variants segregated with the phenotype in this family and matched to the inheritance patterns of autosomal dominant with incomplete penetrance. The number of mutations in PRPF31 and related phenotypes are analyzed using HGMD database (https://www.hgmd.cf.ac.uk/ (accessed on 12 October 2022)) (Appendix A).

The alignment result showed that primers of POS can only be amplified in mutation carriers with PCR product of 301 bp, and the primers of ZC can be amplified in all of them with PCR product of 240 bp (Figure 3). These findings indicated that there was a heterozygous deletion in patients. The deletion position is consistent with the results of WGS.

We then performed the MLPA to further check if there was a large deletion of *PRPF31* exon 1 in affected and unaffected members from this RP family. Through MLPA analysis, we proved that the heterozygous deletion in *PRPF31* exon1 existed in all affected members and three NPCs with a CNV dosage quality (DQ) value in 0.40–0.65 (Figure 4, Appendix A). Additionally, we excluded three possible genes related to RP (including *RHO*, *IMPDH* and *RP1*) out of latent CNV mutation.

After the parametric linkage analysis in, only one region indicated a LOD score > 3.0 suggesting linkage with 24 markers 14 family members, which was located on chr19: 99.03–110.49 (Table 3, Appendix A). Haplotype analysis was then performed in the candidate region. A consistent haplotype was identified in the affected individuals but not in the unaffected individuals (Figure 5).

### 3.3. RNA-Sequencing and Data Analysis

To analyze transcriptional changes among the patient, NPC, and control subject, we performed RNA-sequencing analysis in the family members (III-6, III-7, IV-5, and IV-7). Compared to the healthy control (III-7), RNA sequencing revealed 532 up-regulated genes and 588 down-regulated genes in the patient (III-6) and 24 up-regulated genes and 29 down-regulated genes in NPCs (IV-5 and IV-7). While compared to NPCs, 40 up-regulated genes and 42 down-regulated genes in the patient (III-6) (Table 4, Figure 6). The expression levels of genes in the deletion region and linkage region were listed in Appendix A. We found the expression of *PRPF31* was significantly different between mutation carriers and healthy individual (padj < 0.05), but showed no difference between patient and NPCs (padj > 0.05). GO and KEGG analysis suggested that the main gene location and signal pathway pointed to immunological response and inflammatory (Appendix A).

## 4. Discussion

In this study, we recruited 11 patients and 16 healthy controls from the largest RP11 pedigree with large *PRPF31* deletion. As a classical inherited retinal dystrophy, RP is characterized with striking phenotypic and genetic heterogeneity. Only about 50–70% of RP patients have confirmed genetic defect [1,21]. This is the first time to report the largest adRP pedigree with a large deletion in *PRPF31*, which will provide more genotype–phenotype information for further investigation of RP11.

The patients of this pedigree presented with characteristic RP symptoms, such as night blindness at early ages, visual changes accompanied with characteristic RP-fundus appearance. Though the youngest patient (V-3) with night blindness did not show a significant change in pigment, loss of peripheral PRC outer segment and RPE atrophy were identified in his macular OCT. These finding provide strong evidence for his RP diagnosis. All patients in this pedigree manifested night blindness at about 3 years old. The early onset in some *PRPF31*-related RP patients has not yet been well explained.

### 4.1. Discovery of Pathogenic Mutation in the Pedigree

Due to the limitation of NGS, the pathogenic variant was not detected through panel-based NGS in an initial screening even including *PRPF31* gene. One possible reason is that the length of exon 1 in *PRPF31* (more than 300 bp) is out of range for the pathogenic variant to be read based on NGS; the other possibility is that the deletion is located in an intron region where the primers of panel-based NGS do not capture.

Our WGS analysis identified a large region locus at chr19: 54048499–54118055 with a high LOD score in linkage analysis, harboring *PRPF31*, *VSTM1*, *TARM1*, *OSCAR*, *NDUFA3* and *TFPT*. Furthermore, the deletion positions point at the intron 1 of *VSTM1* and intron 1 of *PRPF31*. The region showed a high LOD score value in linkage analysis.

The reason why large sequence adjacent to *PRPF31* seems more likely to be deleted is that there is a high density of homologous Alu repeats in chr19 [22]. Chromosome 19 is the richest area in Alu repeats associated with the density of GC, which is biologically active [22]. Alu repeats were found to be related to the deleted position [23]. Alu is defined as a class of retroelements termed SINEs (short interspersed elements). It is the main reason for polyadenylation, splicing, and adenosine deaminase that acts on RNA editing [24]. With the evolution of Alu elements in genomic variants, different subfamilies of Alu elements were discovered and classified. The deletion breakpoints identified in this study are both within Alu repeats, adding novel evidence to the literature [25,26]. The nearest Alu repeats adjacent to the deletion are AluJb (chr19: 54048108–54048414) and AluSx1 (chr19: 54117565–54117872). J subfamily is the earliest Alu elements and S subfamily is a really active one [27].

In this pedigree, *PRPF31* is the most likely candidate gene to explain the phenotype of adRP among the 6 genes. Additionally, members with mutated *PRPF31* showed typical incomplete penetrance phenotype, which is supported by the haploinsufficiency theory. Several modifier sequences have been reported to influence expression of *PRPF31*, such as *CNOT3* and *MSR1* [16,28,29,30,31,32,33]. *CNOT3* encodes a subunit of the Carbon Catabolite Repression-Negative On TATA-less (CCR4-NOT) transcriptional complex, and MSR1 is minisatellite element 1 in the promoter of *PRPF31*. Whether other potential regulators could influence expression of *PRPF31* expression in this pedigree is worthy of further exploration.

### 4.2. Gene Expression of PRPF31 and Potential Mechanisms of RP

As reported by previous studies, the *PRPF31* mRNA expression level measured from peripheral blood sample indicated that the patient had a higher level of *PRPF31* expression than healthy control [34]. Our findings suggest that pathogenic variant in *PRPF31* exon 1 could decrease *PRPF31* expression, supporting the haploinsufficiency hypothesis. The *PRPF31* mRNA level from peripheral blood mononuclear cells (PBMCs) of NPC showed no significant difference from the patient in this study, consistent with results in previous studies on leukocytes, fibroblasts and retina organoids [32,34]. Additionally, the expression of *PRPF31* from retina and RPE has been found to be a little higher than the expression from fibroblast [18,32], which shows comparability and variability of the *PRPF31*. This difference in *PRPF31* expression may be influenced by regulators, age, and specific expression level in different tissues. However, RP11 RPE displayed a slightly lower mRNA level and more RP-related shortening of primary cilia compared with NPCs, which indicates there may be a threshold in retina for manifestation of RP phenotypes [32]. Further investigation on expression level of PRPF31 in retina is necessary to explain the incomplete penetrance.

In addition, it is worth determining if other genes in the deletion region will result in the acceleration of RP development and finally induce the early onset in patients in this pedigree. So far, there is no study about the level of transcriptome of genes adjacent to *PRPF31*. We provide meaningful evidence to show the mRNA expression of genes in a large deletion region adjacent to *PRPF31* by RNA-seq. The expression of genes in this region except for *TARM1* showed a reduction in NPCs compared to patient and healthy control, especially for *VSTM1*, *NDUFA3*, *OSCAR* and *TFPT*. The non-penetrance of the variant in carriers may be related to impairing the energy metabolism, inflammatory and apoptosis.

*TFPT (FB1)*, the nearest one adjacent to *PRPF31*, is reported to be related to development of leukemia [35,36], acting as a molecular fusion partner of *TCF3 (E2A)*. *TFPT* was discovered to be connected with the cell cycle [37], induction of programmed cell death (PCD) and proliferation by promoting caspase 9-dependent apoptosis [38]. Furthermore, there was a study on a rat model proving *TFPT* had an important role in modulating cerebral apoptosis [39]. *TCF3* is a kind of transcription factor joining at cellular apoptosis by up-regulating p21 expression and promoting the downstream response of p53 [40]. *TFPT* shares the exon 1 with *PRPF31*, suggesting the existence of a head-to-head structure on DNA strands. So, the heterozygous non-allelic mutation of *TFPT* may induce the apoptosis in retina by influencing the cell cycle and co-regulation with *TCF3 (E2A)*. *TFPT* is related to caspase (3, 7 and 9) apoptosis and PCD. Caspase (3, 7 and 9)-dependent apoptosis, one of PCD, is considered to correlate with rapid photoreceptor degeneration [41]. An RP patient has been reported to carry mitochondrial mutation and show isolated complex I deficiency [42].

*NDUFA3*, the gene in the upstream of *TFPT*, codes for ubiquinone oxidoreductase (complex Ⅰ), compounding the first enzyme in the electron transport chain of mitochondria [43]. *NDUFA3* was reported to be linked with telomere length and aging [44], and it may be a necessary component for stability of transferring Q module in the peripheral arm of the complex Ⅰ [45]. Variants related to Complex Ⅰ such as *NDUFS8*, *NDUFS7*, and *NDUFA2* are found to be explanations for Leigh syndrome, a neurodegeneration showing degenerative foci in the central neural system [46]. Down regulation of the family of NDUF in cortex can promote the energy metabolism disorder in rat brain [47]. With regard to the electron transport function related to *NDUFA3*, it has been believed that mitochondrial dysfunction and metabolism can play distinct—and indispensable—roles in RP development [48,49].

Except for the apoptosis and metabolism stated above, inflammation response may be involved in RP as well. The neural parenchyma of the retina is populated almost solely by microglia, and other immune cells seem almost not to exist in the retina. The recruitment of PBMCs, phagocytes, and microglia in retina can arouse inflammation and phagocytosis so that the development of RP and AMD may be promoted [50,51]. Furthermore, the peripheral inflammatory monocytes are effector cells that mediate cone cell death in RP [52]. Both systemic inflammation and local inflammation are involved in progression of the degenerative retinal diseases [41]. *VSTM-1*, *TARM-1*, and *OSCAR* are related to inflammation in PBMCs. *VSTM1* encodes V-set and transmembrane domain-containing protein 1, a signal inhibitory receptor on leukocytes 1 (SIRL-1), which can negatively regulate oxidative burst in phagocytes [53,54]. *VSTM1* was recognized to be related to inflammation, apoptosis, and necrosis in PBMCs [55,56]. *TARM1* encodes T cell-interacting activating receptor in myeloid cells 1, an arginine residue-containing protein that primarily expressed by monocytes and neutrophils [57]. *TARM1* can be up-regulated in monocytes, macrophages, activated neutrophils, and dendritic cells (DCs) in some inflammatory conditions, and activate the receptor of osteoclast-associated receptor (*OSCAR*) [57]. *TARM1* was located close to *VSTM1*, so they may be a pair of co-receptors that acquired antithetical functions in terms of cellular activation. As for *OSCAR*, it encodes osteoclast-associated receptor, which was a member of the leukocyte receptor complex family. The cell line which it is differentiated from is the same as that of monocytes and DCs. *OSCAR* also can strengthen the inflammatory response and release of cytokine [58,59]. *OSCAR* and *TARM1* share sequence homology and both are activated in the presence of TLR ligands.

In addition, the reduction in *PRPF31* can alter exon usage of genes involved in pre-mRNA splicing (*PRPF3*, *PRPF8*, *PRPF4*, and *PRPF19*) [60]. In order to explore whether there exists an influence of other pre-mRNAs splicing genes, we performed RNA-seq but found the genes related to the process of pre-mRNA splicing (*SNRPD3*, *PHF5A*, *PRPF6*, *SF3A2*, *PRPF3*, *LSM8*, *SNRPA1*, *TXNL4A*, *SNRNP200*, *SART1*) or the components of U4/U6 small nuclear ribonucleoprotein (snRNP) subunits of the spliceosome (*SNU13*, *PRPF3*, *SNRNP27*, *PRPF4*, *USP39*, *SART1*) showed no significantly different expression, except for *SART1*, *LSM8*, *SF3A2*. This may indicate that another potential regulating pathway of RP or different expression pattern in blood sample compared to retina.

### 4.3. Incomplete Penetrance of RP

We sought to elucidate the influential factors of penetrance RP11 in this pedigree using an RNA-seq analysis. Genes in this linkage region (*TNNT1*, *LILRB3*, *RFPL4A*, *KIR2DL1*, *KIR2DL3*, *KIR3DL1*, *KIR3DL2*, *CACNG6*, *TMC4*, *ZNF761*, *ZNF550*) were down-regulated or up-regulated in patient compared to the NPCs or healthy control. *TNNT1* is recognized as a novel marker of immortalized RPE [61], which is known as skeletal muscle-specific troponin. *LILRB3* is associated with the leukocyte immunoglobulin-like receptor. *RFPL4A* encodes the ret finger protein, which is predicted to enable ubiquitin-protein transferase activity and regulation of transcription. *KIR2DL1*, *KIR2DL3*, *KIR3DL1*, and *KIR3DL2* encode killer cell immunoglobulin-like receptor. *CACNG6* is engaged in forming alpha-1 subunits of voltage-gated calcium channel. *TMC4* encodes transmembrane channel like 4. *ZNF761* and *ZNF550* are zinc finger proteins, known for combining with target structure to regulate the expression of gene and cell differentiation. The functions of genes in the linkage region are related to immunology, transmembrane channel, or protein involved in regulation of gene expression. Among them, genes engaged in regulation of transcription were found to be down-regulated in the patient by RNA-seq. It indicated splicing process influenced by *PRPF31* may results in transcription pathways these genes participated in. Based on the GO and KEGG analysis of RNA-seq data, we can figure out the function enrichment is mainly focused on the immunology systems and inflammatory response. Compared to the patient, the immune response and inflammatory apoptotic process were differently regulated in NPCs and healthy control. It provides an indication to guide us to explore the potential mechanism and influence factors from aspects stated above.

For this reason, genes associated with pre-mRNA splicing process, immunology response and inflammation were selected from the differential expression gene list, and those considered as the most relative genes with our study were selected for further research in order to explain the mechanism of incomplete penetrance (Appendix A). We additionally chose *TNNT1*, *GSTM1*, *GSTM3*, *SLIT1*, *IRF7*, *MARCKS*, and *ERAP2* as the potentially relevant gene because of their significantly differential expression and some selection criteria. In conclusion, those genes screened out for further study were based on the principle of its close association with immunology, oxidative stress, ophthalmological disease, and metabolism. *GSTM1* and *GSTM3* are related to glutathione S-transferase Mu, acting as an important factor in redox metabolism [62,63]. *SLPI* is involved in Th2 immune responses and allergic diseases [64]. *IRF7* is a potent regulator of interferon, which shows an effect on antiviral immunity [65]. *ERAP2* is associated with Birdshot uveitis by influencing the immunopeptidome across HLA class Ⅰ allotypes [66]. *MARCKS* served as a marker of diacylglycerol-activation and expressed highly in developmental processes of retina [67,68].

There is an interesting finding that the proband was diagnosed with RP and POAG, which prompted a correlation with these two diseases. Some findings were proposed relating to the potential pathogenic gene. *GSTM1* has been reported as the potential gene involved in open-angle glaucoma [69]. Additionally, both *PRPF8* and *PRPF31* are the members of pre-mRNA processing factor related to RP. *PRPF8* was found to be genotype–phenotype correlation with POAG [70]. It has been hypothesized that the mutation at the C-terminus was associated with RP and the mutation at the N-terminus was associated with POAG. The manifestation of POAG in the proband may be explained by the higher gene expression related to POAG or shown as another phenotype of *PRPF31*, which may lead to POAG and adRP by influencing splicing process in the eye like *PRPF8*.

In summary, we have identified a large deletion region (69 kb) in a five-generation adRP pedigree, in which *PRPF31* was considered as the pathogenic gene. However, we were not able to investigate this gene at the proteome level for further explanation on regulating way of incomplete penetrance in RP. Now we are working on designing a cell model for further study to answer these questions. Based on the iPSC technology, we can establish a stem cell model with a large deletion, which is difficult if using the traditional gene editing method. In addition, we can preserve the intracellular environment of the derived model, so that it will be possible to comprehensively and exactly revalidate the differently expressive gene at the proteomic and transcriptomic levels. Inducing iPSC differentiation into retinal cell will be helpful to elucidate the correlations between the six genes identified in this study and RP phenotype in specific cells types, and to reveal the potential mechanism of incomplete penetrance in RP.

## Figures and Tables

**Figure 1 jcm-11-06682-f001:**
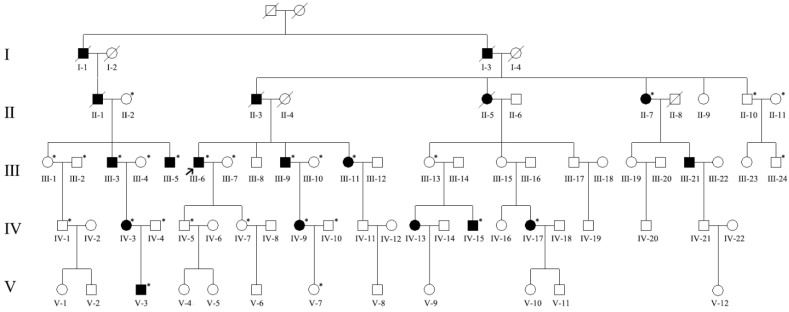
Pedigree of a five-generation family with autosomal dominant retinitis pigmentosa (adRP). Filled symbols represent affected individuals, whereas empty symbols represent unaffected, proband is marked with an arrow. Circles represent females, squares represent males. * represents the member with result of whole-genome sequencing (WGS).

**Figure 2 jcm-11-06682-f002:**
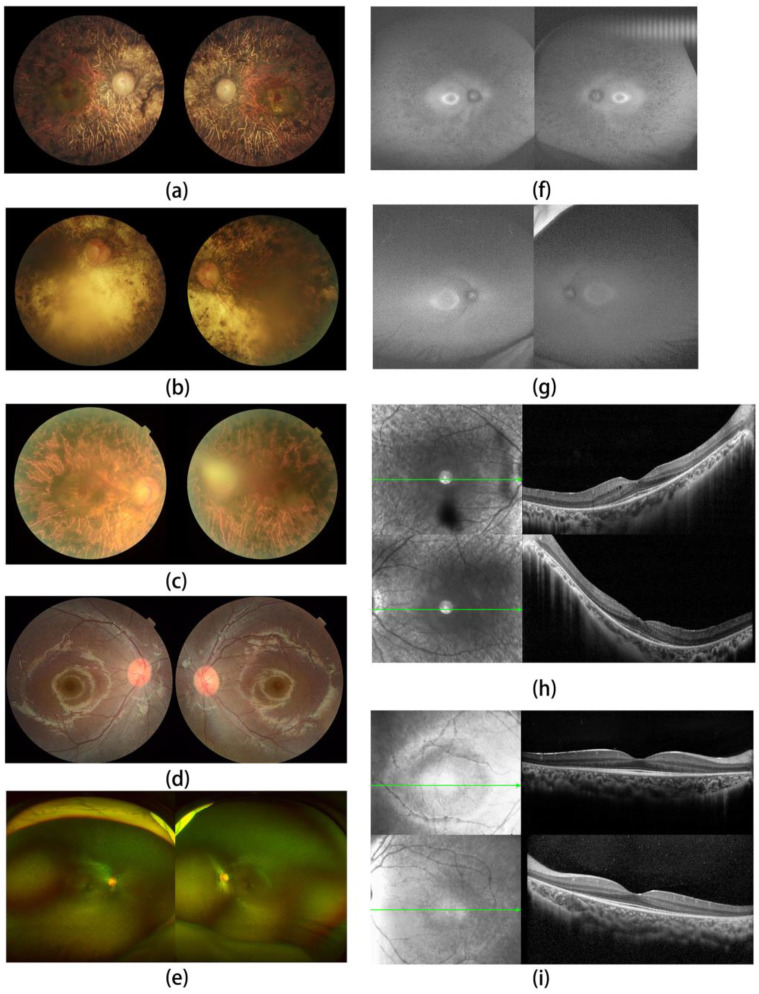
Clinical fundus images. Fundus images presented bone spicules such as pigmentation, constant attenuation of retinal blood vessels and waxy pallor optic disc on the retina in (**a**) III-6, (**b**) III-5, (**c**) III-9 and (**d**) V-3. Full-field fundus images showed lightly retinal pathological change in (**e**) V-3. Fundus autofluorescence (FAF) imaging showed small ring with increase autofluorescence surrounding fovea and area out of the ring with decreased autofluorescenece, which indicated the RPE atrophy in the mid-periphery in (**f**) IV-3; bull’s-eye-like area with increased autofluorescence surrounding fovea can be scanned in the FAF of (**g**) V-3. Optical coherence tomography (OCT) in (**h**) IV-3 presented marked atrophy of the retinal pigment epithelium (RPE), outer retinal tubulation (ORT), and marked thinning of the outer nuclear layers (ONLs), and in (**i**) V-3, it presented loss of the peripheral outer nuclear layer of retina.

**Figure 3 jcm-11-06682-f003:**
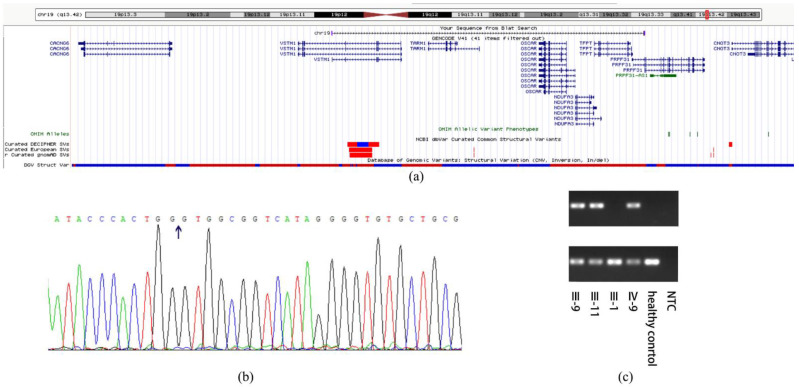
The deletion region and the identification of variant. (**a**) The image of deletion including distribution of genes and comparison to similar variants, the red frame represents structural variants in the region reported by structural variants database listed in the Figure; (**b**) The sequencing results of affected patients. Arrows point at the gap between breakpoints.; (**c**) The first one is 301 bp fragment representing a mutant allele, the second is 240 bp fragment representing a wild-type allele. NTC, no template control.

**Figure 4 jcm-11-06682-f004:**
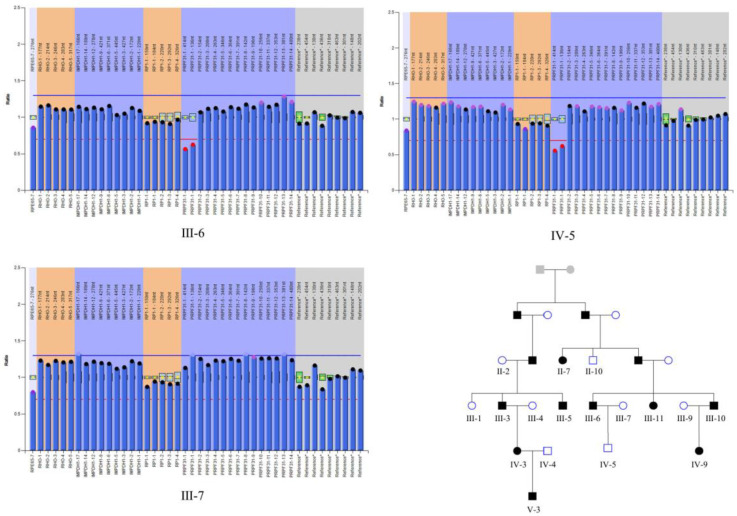
The representative results of multiplex ligation-dependent probe amplification (MLPA) include patients, non-penetrant carriers and healthy controls. Pedigree map of members participating in MLPA. A black symbol indicates that the individual is affected, a white symbol indicates the individual is unaffected, and a grey symbol indicates that the individual’s affectation status is unknown. Squares indicate males and circles indicate females.

**Figure 5 jcm-11-06682-f005:**
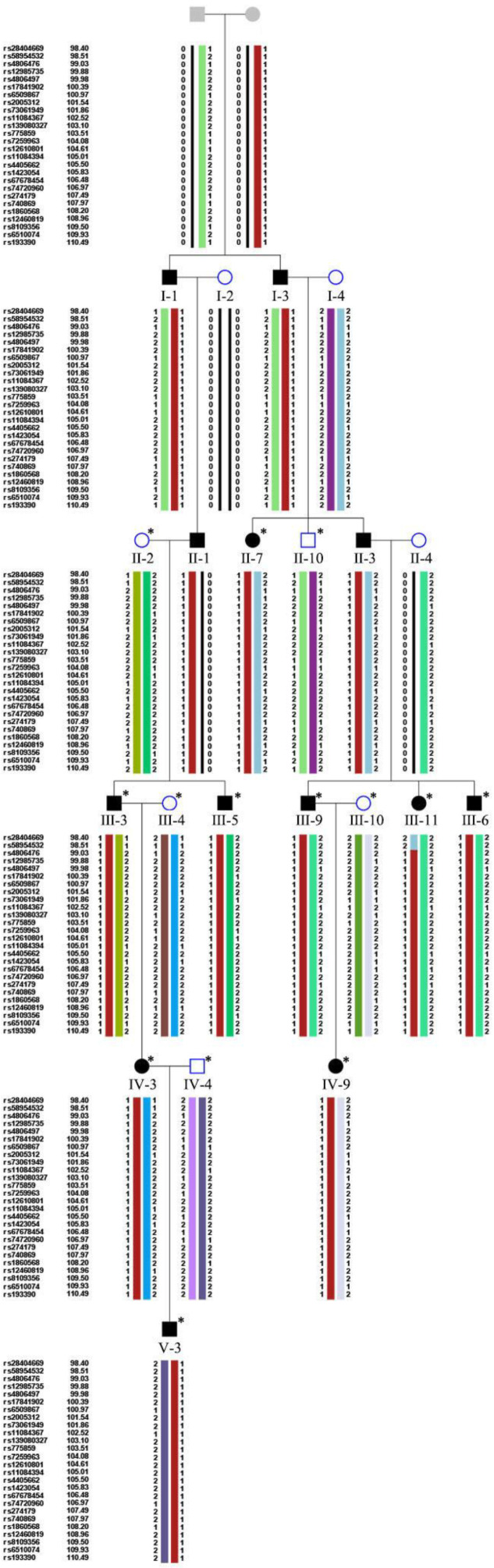
Haplotype analysis. The red columns represent the shared haplotype. Filled symbols, affected individuals; empty symbols, unaffected individuals; grey symbols, individuals in unknown status. SNP markers and their order are described in left side. * represents members using the sequencing data to analyze.

**Figure 6 jcm-11-06682-f006:**
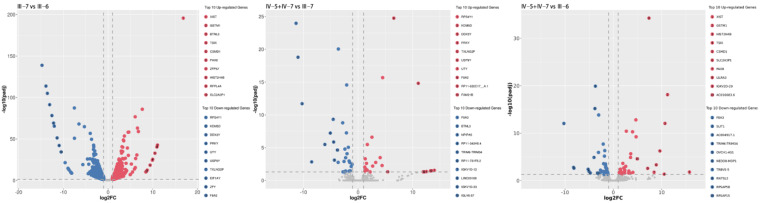
Volcano plot of the significantly differential expressive genes (DEGs) in the core family. *x*-axis, log2FC; y-axis, the negative log10FDR (adjusted *p*-value). Only the top 10 differentially expressed genes are shown in the legend.

**Table 1 jcm-11-06682-t001:** Phenotypes of the RP family members.

Pedigree ID	Age at Examination	Sex	Phenotype	BCVA	IOP (mmHg)	Cup-Disc Ratio	Avg_Axial (mm)	Avg_ACD (mm)	Avg_lens (mm)
II-2	83	Female	Normal	4.7/4.5	16/14	0.3/0.3	21.86/21.75	2.92/2.86	4.69/4.57
II-7	78	Female	RP	HM/LP	13/14	0.4/0.4	21.58/21.43	3.64/2.74	4.94/4.65
II-10	72	Male	Normal	5.3/5.2	15/16	0.4/0.4	22.54/22.44	2.88/2.76	4.74/4.77
II-11	68	Female	Normal	4.7/4.8	18/20	0.3/0.3	21.54/21.64	2.64/2.72	4.72/4.75
III-1	58	Female	Normal	5.2/5.3	15/18	0.4/0.4	22.98/23.08	3.08/3.22	4.57/4.22
III-2	63	Male	Normal	4.9/4.9	19/20	0.6/0.4	22.61/22.47	2.92/2.77	4.76/4.91
III-3	56	Male	RP	-	7/10	0.3/0.3	23.46/23.37	2.24/2.14	5.30/5.33
III-5	54	Male	RP	-	9/10	0.3/0.3	25.77/25.70	2.83/2.72	4.61/4.69
III-6	72	Male	RP	Fc20/HM	18/17	0.9/1.0	22.95/23.51	4.10/4.15	IOL/IOL
III-7	70	Female	Normal	4.7/4.8	14/13	0.4/0.4	22.55/22.66	2.59/2.49	4.68/4.71
III-9	62	Male	RP	4.1/Fc150	22/13	0.6/0.5	24.23/24.07	4.83/3.22	5.38/4.11
III-10	61	Female	Normal	4.8/4.7	11/14	0.3/0.3	21.98/22.12	2.32/2.50	4.48/5.15
III-11	56	Female	RP	4.8/4.7	19/14	0.3/0.3	21.50/22.08	2.71/2.87	3.80/4.23
III-13	61	Female	Normal	5.0/5.0	15/15	0.4/0.4	22.18/22.23	2.68/2.47	4.31/4.43
III-24	45	Male	Normal	5.2/5.1	21/18	0.3/0.3	22.83/22.76	3.02/3.03	4.1/4.05
IV-1	35	Male	Normal	5.2/5.2	19/19	0.3/0.3	24.75/24.66	3.46/3.22	4.15/4.20
IV-7	46	Female	Normal	5.2/5.2	17/18	0.4/0.4	22.83/22.95	3.13/3.36	4.02/3.94
IV-10	47	Male	Normal	5.1/5.1	19/17	0.3/0.3	21.46/21.54	2.86/2.84	4.66/4.65
IV-15	28	Male	RP	4.7/4.7	15/16	0.3/0.3	26.23/24.03	2.57/2.43	4.46/4.55
IV-17	30	Female	RP	5.0/5.0	19/15	0.3/0.3	21.76/21.35	2.83/2.69	4.09/4.10
V-3	4	Female	RP	-	-	0.3/0.3	21.63/21.58	2.65/2.63	3.60/3.58
V-7	16	Female	Normal	5.2/5.3	18/21	0.3/0.3	23.18/23.92	3.18/3.63	3.67/3.61

BCVA, best-corrected visual acuity; IOP, intro ocular pressure; Avg_Axial, average axial length; Avg_ACD, average anterior chamber depth; Avg_lens, average lens thickness. The results were stated as OD/OS; OD, right eye; OS, left eye. RP, retinitis pigmentosa; Fc, counting fingers; HM, hand moving; LP, light perception.

**Table 2 jcm-11-06682-t002:** Results of WGS.

Pedigree ID	Str	End	Chr	SVTYPE	Variant Type
II-2	-	-	-	-	-
II-7	54048499	54118055	19	DEL	Het
II-10	-	-	-	-	-
II-11	-	-	-	-	-
III-1	-	-	-	-	-
III-2	-	-	-	-	-
III-3	54048499	54118055	19	DEL	Het
III-4	-	-	-	-	-
III-5	54048499	54118055	19	DEL	Het
III-6	54048499	54118055	19	DEL	Het
III-7	-	-	-	-	-
III-9	54048499	54118055	19	DEL	Het
III-10	-	-	-	-	-
III-11	54048499	54118055	19	DEL	Het
III-13	54048499	54118055	19	DEL	Het
III-24	-	-	-	-	-
IV-1	-	-	-	-	-
IV-3	54048499	54118055	19	DEL	Het
IV-4	-	-	-	-	-
IV-5	54048499	54118055	19	DEL	Het
IV-7	54048499	54118055	19	DEL	Het
IV-9	54048499	54118055	19	DEL	Het
IV-10	-	-	-	-	-
IV-15	54048499	54118055	19	DEL	Het
IV-17	54048499	54118055	19	DEL	Het
V-3	54048499	54118055	19	DEL	Het
V-7	-	-	-	-	-

SVTYPE, structure variation type; Chr, chromosome; Het, heterozygous variant.

**Table 3 jcm-11-06682-t003:** LOD score of linkage region.

CHR	POS	LABEL	MODEL	LOD	HLOD
19	0.9903	rs4806476	dominant_autosomal	3.1885	3.1885
19	0.9988	rs12985735	dominant_autosomal	3.6074	3.6074
19	0.9998	rs4806497	dominant_autosomal	3.6096	3.6096
19	1.0039	rs17841902	dominant_autosomal	3.6095	3.6095
19	1.0097	rs6509867	dominant_autosomal	3.6093	3.6093
19	1.0154	rs2005312	dominant_autosomal	3.6093	3.6093
19	1.0186	rs73061949	dominant_autosomal	3.6093	3.6093
19	1.0252	rs11084367	dominant_autosomal	3.6092	3.6092
19	1.031	rs139080327	dominant_autosomal	3.6093	3.6093
19	1.0351	rs775859	dominant_autosomal	3.609	3.609
19	1.0408	rs7259963	dominant_autosomal	3.609	3.609
19	1.0461	rs12610801	dominant_autosomal	3.6093	3.6093
19	1.0501	rs11084394	dominant_autosomal	3.6094	3.6094
19	1.055	rs4405662	dominant_autosomal	3.6094	3.6094
19	1.0583	rs1423054	dominant_autosomal	3.6094	3.6094
19	1.0648	rs67678454	dominant_autosomal	3.6093	3.6093
19	1.0697	rs74720960	dominant_autosomal	3.6093	3.6093
19	1.0749	rs274179	dominant_autosomal	3.6091	3.6091
19	1.0797	rs740869	dominant_autosomal	3.6091	3.6091
19	1.082	rs1860568	dominant_autosomal	3.609	3.609
19	1.0896	rs12460819	dominant_autosomal	3.6078	3.6078
19	1.095	rs8109356	dominant_autosomal	3.6066	3.6066
19	1.0993	rs6510074	dominant_autosomal	3.6043	3.6043
19	1.1049	rs193390	dominant_autosomal	3.5849	3.5849

CHR, chromosome; POS, position in the chromosome; LABEL, reference single nucleotide variants; LOD, logarithm of odds score; HLOD, heterogeneity logarithm of odds score.

**Table 4 jcm-11-06682-t004:** The number of differentially expressed genes in RNA-seq analysis.

Group	Up-Regulated	Down-Regulated
Patient vs. healthy control	588	532
NPCs vs. healthy control	24	29
NPCs vs. patient	42	40

Patient, III-6; healthy control, III-7; NPCs, IV-5 and IV-7.

## Data Availability

The data presented in this study are available on request from the corresponding author.

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
