# Peer review of "A 69 kb Deletion in chr19q13.42 including PRPF31 Gene in a Chinese Family Affected with Autosomal Dominant Retinitis Pigmentosa"

_jcm, 2022, doi:10.3390/jcm11226682_

Round 1

Reviewer 1 Report

A study by Yuanzheng Lanet al.  entitled “ A 69-kb deletion with PRPF31 in chr19q13.42 in a Chinese autosomal dominant retinitis pigmentosa family” presents a large family of 5 generations with a history of retinal degeneration.  The authors identified a novel mutation in pedigree, causative of PRPF31-related adRP and proposed the mechanism of action by thorough ophthalmic phenotyping and genotyping using a wide range of genetic methods such as WGS, MLPA, linkage analysis and haplotype, RNA-seq.

This is broadly known that severity of PRPF31-related RP is variable in different members of the same family, and it depends on the type of mutant allele inherited, the level at which this allele is expressed, and the level at which the wild-type allele is expressed. Furthermore, obligate carriers may be totally asymptomatic, showing complete non-penetrance. Usually, large insertions or deletions are predicted to lead to complete loss of protein expression, however this is not always the case.

This study has few main points:

1.     This manuscripts presents ophthalmic phenotype and genotype of the studied family using broad range of genetic technologies. Was an ERG performed in the studied family?

2.     Figures and tables are presented in clear and readable way.

Weaknesses:

1.     Phrasing/language edits require major improvement through the entire manuscript. I would suggest editing by English-language professional. Due to linguistic issues, it is difficult to understand the meaning of the message the authors wish to deliver in many parts.

2.     Through the entire manuscript (Abstract, Introduction and Results section) text does not flow well, requires better structure and coherence. Material and Methods are the best written, only some improvements are required.

Broad comments:

1.     Consider changing the title to: “A 69-kb deletion in chr19q13.42 including PRPF31 gene in a Chinese family affected with autosomal dominant retinitis pigmentosa”.

2.     Please, re-write the abstract and introduction for better coherency and language (see specific comments below).

3.     Discussion section: I would suggest to make subtitles. Somehow, this section is very long. I propose shortening the text and connect the paragraphs in a better way and being more specific. It is not easy to pick the main message of this article as for now.

4.     Please consider changing “mutation” to “pathogenic variant” in the entire manuscript.

5.     Make sure that nomenclature of variants is presented according to the most recent recommendations of HGVS.

Specific comments:

1.     Abstract, row 16: change to “We aimed to identify a cause of autosomal dominant retinitis pigmentosa and characterize the underlying molecular mechanisms of incomplete penetrance in Chinese family affected with adRP”.

2.     Abstract, row 18: change to “Recruited family members underwent…”

3.     Abstract, row 20-21: change to “were performed to identify the genetic cause and mechanism of action…”

4.     Abstract, row 21: Please, re-write the entire statement “In an incomplete penetrance core family,….”

5.     Row 22: change to “In the studied family, 14 individuals were found to have a novel heterozygous large deletion….”. In general rows 22-31 should be re-organised with both improved language and better structure. It is now a bit chaotic.

6.     Introduction, row 35: change to “Retinitis pigmentosa (RP) is one of the most common forms of inherited retinal degenerations (IRDs) and affects 1:4000 worldwide (ref.). This disease affects predominantly rod photoreceptor cells with later involvement of cone photoreceptor cells. Phenotypic and genotypic heterogeneity are common. Night blindness is usually the first symptom….”

7.     Introduction, rows 41-43: improve this statement “Non-mendelian inheritance patterns including digenic inheritance and maternal (mitochondrial) inheritance are also known in RP…”

8.     Introduction, row 44: change to “related to”

9.     Introduction, row 47: statement is unclear. Change to “Until now, there is not known therapy for PRPF31-related adRP…”

10.  Introduction, rows 47-53: need to be edited for improved language and flow.

11.  Introduction, row 57-59: improve this statement as “The other genes…..restricted to retina.”

12.  Introduction, row 59-61: merge two sentences into one. Change “reported as pathologic gene for RP11” to “Pathogenic variants in PRPF31 have been reported to be causative of ad RP...”

13.  Introduction, row 61-62: merge two sentences “our body. But mutant…” as e.g. “our body, however mutant…”

14.  Introduction, row 64: “another significant phenotype of PRPF31 is incomplete penetrance…” – change this statement, this is very unclear. Incomplete penetrance is not a phenotype, but milder phenotype is due to incomplete penetrance.

15.  Introduction, row 69: change phrasing “the phenotype related to RP11 may not happen”

16.  Introduction, row 65-66: please change this sentence to e.g., “In our study, we present a 5-generation family with early-onset RP with autosomal dominant inheritance mode…”

17.  Introduction, row 77-78: remove  “What’s more”, use “Furthermore” and improve this sentence linguistically.

18.  Introduction, row 79: remove a word “some”. In general this statement requires improvement of English language, please change phrasing of this “, to figure out what is engaged in the onset of RP.”

19.  Introduction, rows 81-84: correct the entire statement, phrasing. It is unclear.

20.  Materials and Methods, section Participants: major English language improvements are required.

21.  Materials and Methods, Whole genome sequencing section: improve first sentence- does it mean that family didn’t have an initial molecular diagnosis before enrolment to this study?

22.  Results section, rows 192-194: is this statement necessary?

23.  Results, rows 196-197: start this sentence “We obtained a pedigree of family…”

24.  Results, rows 198-200: change phrasing in this sentence. What means “special phenotype”? Is it not better to write here that individuals in the studied family had symptoms, such as….

25.  Results, row 200-201: change the sentence to “The fundus exam revealed….”

26.  Results, row 203: please do not start the sentence with “And”

27.  Results, rows 203-207: start this sentence as “In the youngest patient, who already presented with night blindness, we found…” and here describe imaging findings.

28.  Results, rows 207-211: please, edit this sentence (grammar and coherence).

29.  On Figure 2: write findings in the Figure legend and remove the detailed description of the findings from the Results section accordingly.

30.  Table 1: add that BCVA was in logMAR.

31.   Results, rows 225-228: start the sentence with “We were not able to find…”. What means “fundus disease panel”? Do the authors mean “IRD panel”? Write how many genes were included.

32.  Results, row 231-231: do not start the sentence with “And”, and “What’s more,…”

33.  Results, row 237: change to “we focused analysis on PRPF31 as a candidate gene…”

34.  Results, row 244: remove “normal people” and write “healthy individuals used as controls in our study…”

35.  Results, rows 247-248: change to “This finding indicated that there was a heterozygous deletion present in 3 patients…”

36.  Discussion, row 297: change to “healthy controls”.

37.  Discussion, row 299-300: improve language in this sentence.

38.  Discussion, rows 301-302: change to e.g., “The patients of this pedigree presented with characteristics RP symptoms, such as  night blindness in the early age, visual changes accompanied with characteristic RP-fundus appearance”.

39.  Discussion, rows 305-309: please, improve this section for easier readability.

40.  Discussion, rows 310-311: change to e.g.,  “Due to limitation of NGS, the pathogenic variant was not detected through panel testing even including PRPF31 gene”.

41.  Discussion, rows 311-314: merge two sentences and start the sentence with “One of the reason is that…., the other reason is that the position of the deletion is….”

42.  Discussion, row 317: exchange “And” with “Furthermore,…” as a start of this sentence.

43.  Discussion, rows 318-320: suggest to place this sentence in the first paragraph of the Discussion section.

44.  Discussion, row 336: change “Meanwhile” to other phrasing.

45.  Discussion, rows 337: remove “now” from the sentence.

46.  Discussion, rows 337-344: please improve structure of this section.

47.  Discussion, rows 348: change “But” to another word at the beginning of this sentence.

48.  Discussion, rows 358-360: please improve phrasing and coherence of this text.

49.  Discussion, rows 366-373: change phrasing in this section and language for better readability.

50.  Discussion, rows 392-420: these four paragraphs lack connection among one another. They appear as separate texts.

51.  Discussion, row 430: change phrasing in this sentence.

52.  Discussion, row 469: change to “ as the potential gene involved..”

53.  Discussion, last paragraph: start with “In summary…”

54.  Discussion, row 481: change this phrasing “in pathogenic process…”

55.  Discussion, rows 283-490: there are future directions, add subtitle here. Improve phrasing in this section.

Author Response

Dear Reviewer:

Thank you for your comments. Those comments are all valuable and very helpful for revising and improving our paper, as well as the important guiding significance to our research. We have studied comments carefully and have made correction which we hope meet with approval. Revised version is marked in red in the paper using "Track Changes". The responses we made point-to-point are stated as follows:

"This study has few main points:

  1. This manuscript presents ophthalmic phenotype and genotype of the studied family using broad range of genetic technologies. Was an ERG performed in the studied family?”

ERG was performed in some patients but not all. Considering the limitations of immobility of ERG and difficulty of recruitment, we didn’t specially recruit all members to our hospital for ERG.

Another issue on the Result section is that stated below.

“2. Figures and tables are presented in clear and readable way.”

Thank you for your advice. The Figures may be compressed in the process of edit and submit. To be clearer and in accordance with the reviewer concerns, we have paid attention to this problem and re-edit the figures and tables.

Weaknesses and responses:

  1. Phrasing/language edits require major improvement through the entire manuscript. I would suggest editing by English-language professional. Due to linguistic issues, it is difficult to understand the meaning of the message the authors wish to deliver in many parts.

We are grateful for your suggestions. We will cherish the opportunity to re-write the manuscript. We are regretful for difficulty of understanding caused by the linguistic issues. This manuscript has been revised extensively according to your constructive suggestions. In addition, the expression of the manuscript has been improved with the help of a native English speaker (including Abstract, Introduction, Results section and Material and Methods). We made our efforts to improve the language for better understanding the manuscript.

  1. Through the entire manuscript (Abstract, Introduction and Results section) text does not flow well, requires better structure and coherence. Material and Methods are the best written, only some improvements are required.

Thanks a lot for the comments. We paid attention to the structure and coherence. We re-writed the unclear sentences and tried our best to deliver the meaning.

Broad comments and responses:

  1. Consider changing the title to: “A 69-kb deletion in chr19q13.42 including PRPF31 gene in a Chinese family affected with autosomal dominant retinitis pigmentosa”.

Thank you for the important suggestion. The precedent version of the title has been replaced, becoming “A 69-kb deletion in chr19q13.42 including PRPF31 gene in a Chinese family affected with autosomal dominant retinitis pigmentosa”.

  1. Please, re-write the abstract and introduction for better coherency and language (see specific comments below).

Thank you for your detailed and professional comments. We have made extensive English editing about the abstract, introduction, results and discussion. (The specific responses to comments are stated below.)

  1. Discussion section: I would suggest to make subtitles. Somehow, this section is very long. I propose shortening the text and connect the paragraphs in a better way and being more specific. It is not easy to pick the main message of this article as for now.

We fully agree with the comment and have made subtitles in the Discussion section. (Line 381, 423 and 517)

  1. Please consider changing “mutation” to “pathogenic variant” in the entire manuscript.

Sorry for our faults. The “mutation” has been changed to “pathogenic variant”.

5.Make sure that nomenclature of variants is presented according to the most recent recommendations of HGVS.”

We re-checked the nomenclature of variants and comfirmed that it was presented according to the most recent recommendations of HGVS.

According to the specific comments, our responses to the specific comments are as follows:

Specific comments and responses:

  1. Abstract, row 16: change to “We aimed to identify a cause of autosomal dominant retinitis pigmentosa and characterize the underlying molecular mechanisms of incomplete penetrance in Chinese family affected with adRP”.

The sentence was changed according to the comment. (Line 18-20)

  1. Abstract, row 18: change to “Recruited family members underwent

The sentence was changed to “All enrolled family members underwent”. (Line 21)

  1. Abstract, row 20-21: change to “were performed to identify the genetic cause and mechanism of action…”

We agreed and have updated the sentence. (Line 23)

  1. Abstract, row 21: Please, re-write the entire statement “In an incomplete penetrance core family,….”

The sentence was re-written to make better comprehension. (Line 24)

  1. Row 22: change to “In the studied family, 14 individuals were found to have a novel heterozygous large deletion….”. In general rows 22-31 should be re-organised with both improved language and better structure. It is now a bit chaotic.

The sentence was changed according to the comment. (Line 27-36)

  1. Introduction, row 35: change to “Retinitis pigmentosa (RP) is one of the most common forms of inherited retinal degenerations (IRDs) and affects 1:4000 worldwide (ref.). This disease affects predominantly rod photoreceptor cells with later involvement of cone photoreceptor cells. Phenotypic and genotypic heterogeneity are common. Night blindness is usually the first symptom….”

Thanks a lot for your comment. The sentence has been changed. (Line 46-49)

  1. Introduction, rows 41-43: improve this statement “Non-mendelian inheritance patterns including digenic inheritance and maternal (mitochondrial) inheritance are also known in RP…”

We have revised the text to address your concerns and hope that it is now clearer. (Line 81-83)

  1. Introduction, row 44: change to “related to”

The word was changed to “related to”.

  1. Introduction, row 47: statement is unclear. Change to “Until now, there is not known therapy for PRPF31-related adRP…”

The sentence was changed according to the comment. (Line 64)

  1. Introduction, rows 47-53: need to be edited for improved language and flow.

The section was edited to make understanding easier. (Line 64-70)

  1. Introduction, row 57-59: improve this statement as “The other genes…..restricted to retina.”

The sentence was improved. (Line 81-83)

  1. Introduction, row 59-61: merge two sentences into one. Change “reported as pathologic gene for RP11” to “Pathogenic variants in PRPF31 have been reported to be causative of ad RP...”

We merged two sentences into one according to the comment. (Line 77-78)

  1. Introduction, row 61-62: merge two sentences “our body. But mutant…” as e.g.: “our body, however mutant…”

The two sentences were re-edited. (Line 83-86)

  1. Introduction, row 64: “another significant phenotype of PRPF31 is incomplete penetrance…” - change this statement, this is very unclear. Incomplete penetrance is not a phenotype, but milder phenotype is due to incomplete penetrance.

We have modified this expression throughout the sentence according to the comment. (Line 87)

  1. Introduction, row 69: change phrasing “the phenotype related to RP11 may not happen”

We agreed to it and have re-phrased it. (Line 93-94)

  1. Introduction, row 65-66: please change this sentence to e.g., “In our study, we present a 5-generation family with early-onset RP with autosomal dominant inheritance mode…”

Thanks for your constructive suggestion. The sentence was changed. (Line 101-102)

  1. Introduction, row 77-78: remove “What’s more”, use “Furthermore” and improve this sentence linguistically.

We changed “What’s more” to “Furthermore” and improved the sentence. (Line 105-106)

  1. Introduction, row 79: remove a word “some”. In general, this statement requires improvement of English language, please change phrasing of this “, to figure out what is engaged in the onset of RP.”

We agreed and have removed the sentence because it didn’t cause to misunderstand the text. (Line 107-112)

  1. Introduction, rows 81-84: correct the entire statement, phrasing. It is unclear.

As is stated above, the statement has been removed.

  1. Materials and Methods, section Participants: major English language improvements are required.

We have improved the section. (Line 115-124)

  1. Materials and Methods, Whole genome sequencing section: improve first sentence- does it mean that family didn’t have an initial molecular diagnosis before enrolment to this study?

We have improved the section. (Line 115-124)

  1. Results section, rows 192-194: is this statement necessary?

Sorry for the redundant part. It was removed.

  1. Results, rows 196-197: start this sentence “We obtained a pedigree of family…”

The sentence was re-written. (Line 237)

  1. Results, rows 198-200: change phrasing in this sentence. What means “special phenotype”? Is it not better to write here that individuals in the studied family had symptoms, such as….

We re-phrased the text for better understanding the meaning we want to deliver. (Line 241)

  1. Results, row 200-201: change the sentence to “The fundus exam revealed….”

The sentence was changed according to the comment. (Line 243)

  1. Results, row 203: please do not start the sentence with “And”

We have removed the sentence. The description was added to the figure legend. (Line 266-273)

  1. Results, rows 203-207: start this sentence as “In the youngest patient, who already presented with night blindness, we found…” and here describe imaging findings.

Thanks for the suggestion. The sentence was revised. (Line 250-253)

  1. Results, rows 207-211: please, edit this sentence (grammar and coherence).

We have edited the sentence. (Line 254-259)

  1. On Figure 2: write findings in the Figure legend and remove the detailed description of the findings from the Results section accordingly.

Thanks a lot for the constructive suggestions. We have re-written the Results section and added the description in the Figure legend. (Line 266-273)

  1. Table 1: add that BCVA was in logMAR.

We re-added the BCVA in logMAR. (Line 274)

  1. Results, rows 225-228: start the sentence with “We were not able to find…”. What means “fundus disease panel”? Do the authors mean “IRD panel”? Write how many genes were included.

The sentence was edited according to the comment. We are sorry for the unclear phrase and change it to “IRD panel”. The number of genes is added in the manuscript. (Line 280-282)

  1. Results, row 231-231: do not start the sentence with “And”, and “What’s more,…”

The words were removed. (Line 287)

  1. Results, row 237: change to “we focused analysis on PRPF31 as a candidate gene…”

The sentence was revised according to the comment. (Line 293)

  1. Results, row 244: remove “normal people” and write “healthy individuals used as controls in our study…”

Sorry for the wrong phrase. We think the sentence is not suitable for the Results section. It was removed without loss of comprehension. (Line 301-302)

  1. Results, rows 247-248: change to “This finding indicated that there was a heterozygous deletion present in 3 patients…”

The sentence was changed to “These findings indicated that there was a heterozygous deletion in patients.” (Line 305-306)

  1. Discussion, row 297: change to “healthy controls”.

This phrase was changed according to the comment (Line 362).

  1. Discussion, row 299-300: improve language in this sentence.

The language of the sentence was improved for readability. (Line 365-366)

  1. Discussion, rows 301-302: change to e.g., “The patients of this pedigree presented with characteristics RP symptoms, such as night blindness in the early age, visual changes accompanied with characteristic RP-fundus appearance”.

We have revised the text and hope that it is now clearer. (Line 369-370)

  1. Discussion, rows 305-309: please, improve this section for easier readability.

This section was re-written to make full comprehension of it. (Line 376-380)

  1. Discussion, rows 310-311: change to e.g., “Due to limitation of NGS, the pathogenic variant was not detected through panel testing even including PRPF31 gene”.

This sentence was changed to “Due to limitation of NGS, the pathogenic variant was not detected though panel-based NGS in an initial screening even including PRPF31 gene.” (Line 382-383).

  1. Discussion, rows 311-314: merge two sentences and start the sentence with “One of the reason is that…., the other reason is that the position of the deletion is….”

The two sentences were rephrased and merged according to the comment: “One possible reason is that the length of exon 1 in PRPF31 (more than 300 bp) is out of range in which pathogenic variant can be read based on NGS, the other possibility is that the deletion is located in an intron region where the primers of panel-based NGS do not capture.” (Line 383-388)

  1. Discussion, row 317: exchange “And” with “Furthermore,…” as a start of this sentence.

This word has been instead of “Furthermore”. (Line 392)

  1. Discussion, rows 318-320: suggest to place this sentence in the first paragraph of the Discussion section.

We appreciated and accepted your advice. The sentence was revised and replaced in the first paragraph of the Discussion section. (Line 367-368)

  1. Discussion, row 336: change “Meanwhile” to other phrasing.

This word was changed to “Additionally”. (Line 412)

  1. Discussion, rows 337: remove “now” from the sentence.

This word was removed. (Line 413)

  1. Discussion, rows 337-344: please improve structure of this section.

We have made the changes and hope it is now clearer. (Line 413-422)

  1. Discussion, rows 348: change “But” to another word at the beginning of this sentence.

The word has been removed. (Line 429)

  1. Discussion, rows 358-360: please improve phrasing and coherence of this text.

Modified throughout the text according to the comment (Line 440-443).

  1. Discussion, rows 366-373: change phrasing in this section and language for better readability.

Thank you for the constructive comment. We agree and have revised the text. (Line 449-458)

  1. Discussion, rows 392-420: these four paragraphs lack connection among one another. They appear as separate texts.

It is really true as reviewer suggested. We tried to revise the text for better readability. (Line 477-507)

  1. Discussion, row 430: change phrasing in this sentence.

This sentence was rephrased. (Line 518)

  1. Discussion, row 469: change to “as the potential gene involved..”

It was changed to “as the potential gene for”. (Line 560-561)

  1. Discussion, last paragraph: start with “In summary…”

The phrase “In all” was changed to “In summary”. (Line 570)

  1. Discussion, row 481: change this phrasing “in pathogenic process…”

Considering to the readability of the text, this sentence was removed. (Line 571-574)

  1. Discussion, rows 283-490: there are future directions, add subtitle here. Improve phrasing in this section.

We added subtitles in the Discussion section. (Line 381, 423 and 517)

We tried our best to improve the manuscript and made some changes in the manuscript. These changes will not influence the content and framework of the paper. And we did not list all the changes but marked in red in revised paper using the “Track Changes” function. We appreciate for Editors/Reviewers’ warm work earnestly, and hope that the correction will meet with approval.

Once again, thank you very much for your comments and suggestions.

Sincerely,

The Authors

Reviewer 2 Report

There are some major and minor comments that might help in improving the overall presentation. 

1. Could alternative rare variants contribute to the phenotype seen in these patients?

2. The gene name should be italic throughout the manuscript.

3. The manuscript, especially the clinical and discussion description should be edited by a native English speaker.

4. Please create clinical and molecular investigation tables. Also mention a column in the molecular table that how many mutations (missense, nonsense, frameshift, splice site) are reported in PRPF31 up to date using the HGMD database.

5. Please mention bioinformatics tools with web sources.

6. Figures Quality is very bad. Please make a clean figure and also write the figure legend in scientific English.

7. Please follow the HGVN (http://varnomen.hgvs.org/) for identified mutations.

Author Response

Dear Reviewer,

Thank you for your comments. Those comments are all valuable and very helpful for revising and improving our paper, as well as the important guiding significance to our research. We have studied comments carefully and have made correction which we hope meet with approval. Revised version is marked in red in the paper using "Track Changes". The responses we made point-to-point are stated as follows:

Comments:

“There are some major and minor comments that might help in improving the overall presentation.”

Comment 1:

  1. Could alternative rare variants contribute to the phenotype seen in these patients?

Response 1:

No other possible variants in PRPF31 were found but this huge deletion. And we didn’t find other adRP pathogenic genes in this pedigree. We believed the results are reliable by WGS. We used the Sanger sequencing to verify the variants. The parametric linkage analysis and haplotype were performed to make sure the relevance of phenotype and deletion.

Comment 2:

  1. The gene name should be italic throughout the manuscript.

Response 2:

We are very sorry for our incorrect writings. Thanks a lot for the comment. We have fixed the errors.

Comment 3:

  1. The manuscript, especially the clinical and discussion description should be edited by a native English speaker.

Response 3:

We have re-written the manuscript according to the Reviewer’s suggestion. We have carefully and thoroughly proofread the manuscript to correct all the grammar and typos with the help of a native English speaker.

Comment 4:

  1. Please create clinical and molecular investigation tables. Also mention a column in the molecular table that how many mutations (missense, nonsense, frameshift, splice site) are reported in PRPF31 up to date using the HGMD database.

Response 4:

We are thankful for the constructive suggestion. The clinical and molecular investigation table was added in the supplementary material using the HGMD database. The relevant contents are provided below for your quick reference.

Table S4 The number of mutations in PRPF31 and related phenotypes

Phenotype

Missense/nonsense

Splicing

Small deletions

Small insertions

Small indels

Gross deletions

Gross insertions/duplications

Complex rearrangements

Repeat variants

Regulatory

Retinitis Pigmentosa

62

34

56

25

4

20

7

3

0

0

Retinal disease

5

0

1

0

0

0

0

0

0

0

Retinal Dystrophy

4

2

2

0

0

0

0

0

0

0

Retinal Degeneration

1

1

5

2

0

0

0

0

0

0

Leber congenital amaurosis

1

0

0

0

0

0

0

0

0

0

Development disorder

0

1

0

0

0

0

0

0

0

0

Total

73

38

64

27

4

20

7

3

0

0

Comment 5:

  1. Please mention bioinformatics tools with web sources.

Response 5:

We agree and have added the web sources in the Materials and Methods section.

Comment 6:

6.Figures Quality is very bad. Please make a clean figure and also write the figure legend in scientific English.”

Response 6:

Thanks for your kind suggestion, which is valuable for improving the readability of the manuscript.

Comment 7:

7.Please follow the HGVN (http://varnomen.hgvs.org/) for identified mutations.

Response 7:

We re-checked the nomenclature of variants and confirmed that it was presented according to the most recent recommendations of HGVS.

We would like to take this opportunity to thank you for all your time involved and this great opportunity for us to improve the manuscript. We hope you will find this revised version satisfactory.

Sincerely,

The Authors

Reviewer 3 Report

The manuscript presented by Lan et al. shows a systematic approach to identify and characterize large deletions affecting the PRPF31 gene, which show to be responsible for the onset of adRP in a 5-generation Chinese family with a history of incomplete penetrant RP11.

The work presented starts with the recruitment of several members of this family, including affected and unaffected individuals. Authors report that common NGS techniques are unable to identify the disease-causing mutation and go on to perform WGS, after which they identify a 69-kb deletion region affecting PRPF31 and other neighboring genes such as VSTM-1, TARM-1, OSCAR, NDUFA3 and TFPT. Later, using MLPA and linkage analysis, they confirm the incomplete penetrance of the phenotype and identify SNPs in a candidate region related with the phenotype. Finally, they perform RNA-seq comparing expression levels of genes in control, NPC and affected members of the family, and find a set of differentially expressed genes, mostly related with immune response and inflammation.

The presented study uses state-of-the-art technology to identify and characterize the large deletion present in these patients suffering from RP11, and even proposes an explanation to the incomplete penetrance of the disease phenotype in the family.

Even though the authors describe this study as a first-time report of an adRP pedigree with large deletion in PRPF31, there is a lack of novelty in the work presented. There is a myriad of other studies published in the last 12 years that already reported the presence of large deletions affecting PRPF31 and neighboring genes. As it has been previously reported, PRPF31 lies within a genomic region rich in repeat elements, especially Alu repeats. We now know that large deletions causing RP11 are not rare events, as they are usually caused by Alu-mediated nonallelic homologous recombination.

Other concerns:

·  Language should be improved. There are several typos, misuses of prepositions and pronouns, and redundant sentences, such as the ones on lines 60-63.

·      Line 46: I think the statement saying that more than 8000 genes have been found to be related with RP is wrong. I guess authors meant mutations and not genes.

·      Line 57: Missing reference

·      Please use adRP, arRP and xlRP instead of ADRP, ARRP and XLRP.

·      The use of 2nd generation NGS-based whole genome sequencing can lead to misinterpretations. Genome sequencing of short-read sequences of regions rich in repetitive sequences, such as Alu-repeats, can lead to inaccurate alignment which inhibit accurate alignment. 3rd generation NGS, such as Nanopore sequencing, offer a much better alternative, as they are able to produce long reads up to 2Mb in length.

·      Remove text on lines 192-194

·  Authors describe that affected patients display differential expression of several genes related with immunological response and inflammation. It would be good to complete such discovery with medical examinations of these patients to check if they present metabolic abnormalities or immune deficiencies.

Author Response

Dear Reviewer,

We sincerely thank you for thoroughly examining our manuscript and providing very helpful and encouraging comments to guide our revision. We also hope that our explanation has fully addressed all of your concerns. In the remainder of this letter, we discuss each of your comments individually along with our corresponding responses.

Major comments:

“Even though the authors describe this study as a first-time report of an adRP pedigree with large deletion in PRPF31, there is a lack of novelty in the work presented. There is a myriad of other studies published in the last 12 years that already reported the presence of large deletions affecting PRPF31 and neighboring genes. As it has been previously reported, PRPF31 lies within a genomic region rich in repeat elements, especially Alu repeats. We now know that large deletions causing RP11 are not rare events, as they are usually caused by Alu-mediated nonallelic homologous recombination.”

Responses:

Thanks a lot for the comments. Although there are some reports focusing on large deletion in PRPF31, the focus on the relevance of the early-onset and the huge deletion has been rarely explored. No 5-generation RP11 pedigree presenting early-onset has been reported before. The finding is reported for the first time compared to the previous studies. The early-onset retinitis pigmentosa family, the huge deletion in 19q13.42, the transcriptional level in PRPF31 inconsistent with the clinical phenotype, and the expression of the 6 genes in large deletion may be our novel findings.

Minor Comments:

  1. Language should be improved. There are several typos, misuses of prepositions and pronouns, and redundant sentences, such as the ones on lines 60-63.

Thanks for your constructive suggestion, which is highly appreciated. We have carefully scrutinized the manuscript, and typos, grammatical errors and long sentences, etc.

  1. Line 46: I think the statement saying that more than 8000 genes have been found to be related with RP is wrong. I guess authors meant mutations and not genes.

We are grateful for your kind remind. We were going to write 300 genes here but we made a mistake. The sentence was re-phrased. (Line 62)

  1. Line 57: Missing reference

Sorry for our negligence. In fact, PRPF31 is the second most common gene in most population in some research. We re-wrote the statement and added the reference in the revised manuscript. The references are now included in the revised manuscript. (Line 76)

  1. Please use adRP, arRP and xlRP instead of ADRP, ARRP and XLRP.

We are sorry for the incorrect writings. This observation is correct. We have changed.

5.The use of 2nd generation NGS-based whole genome sequencing can lead to misinterpretations. Genome sequencing of short-read sequences of regions rich in repetitive sequences, such as Alu-repeats, can lead to inaccurate alignment which inhibit accurate alignment. 3rd generation NGS, such as Nanopore sequencing, offer a much better alternative, as they are able to produce long reads up to 2Mb in length.

Nice suggestions! In order to double check the breakpoint region, we used the Sanger sequencing to verify the variants. We believed the results are reliable. Your suggestion indicated us that 3rd generation NGS can be used to sequence out the short-read sequences of regions rich in repetitive sequences in the future work.

  1. Remove text on lines 192-194

We are sorry for our negligence and have already fixed that.

7.Authors describe that affected patients display differential expression of several genes related with immunological response and inflammation. It would be good to complete such discovery with medical examinations of these patients to check if they present metabolic abnormalities or immune deficiencies.

The patients didn’t present any metabolic or immune disease after medical examinations. We suppose that the light changes of immunological response and inflammation won’t cause the related systemic diseases but will affect the retina because of the high demand of metabolism of retina.

We would like to take this opportunity to thank you for all your time involved and this great opportunity for us to improve the manuscript. We hope you will find this revised version satisfactory.

Sincerely,

The Authors

Reviewer 4 Report

Authors describe a pedigree of a family with autosomal dominant retinitis pigmentosa (ADRP) related to mutation in PRPF31. The article appear scientifically sound although reading is quite difficult due to poor English grammar. I suggest a strong revision of the text from a native English speaker because the topic is interesting and worth of consideration: it shows the potential of WGS, although we think that array-CGH would have been a viable option to show the described deletion.

Following some specific points:

Introduction, page 2, line 46: 8000 genes associated to RP is questionable, up to now on RetNet RP is associated to about 300…

Introduction, page 2, line 65: …”may not present” seems more suitable than “…may not appear….”

Results Section, page 5, lines 192-194: please remove the sample paragraph

Results section, 3.1 paragraph, page 5, line 101: “…typical RP signs..” is maybe better than “…typical RP symptoms..”

Results section, 3.1 paragraph, page 5, lines 204-206: “We found bull’s-eye maculopathy in fundus autofluorescence of macular and atrophy of retinal pigment epithelial cells and loss of the peripheral outer nuclear layer of retina in macular OCT in the youngest patient”

Please rephrase “We found bull’s-eye maculopathy in fundus autofluorescence, atrophy of retinal pigment epithelium and loss of the peripheral outer nuclear layer of retina in OCT scan of the youngest patient”: if it is corresponding to the phenotype described it sounds better than the description provided

Same paragraph, line 207: please correct “…have appeared..” with “manifested” or “presented”

For readers it would be useful an explicative image of the deletion to better see which genes contains and the distribution: moreover, there are many CNVs database (i.e. UCSC, decipher) but no information on their use is provided to compare similar deletions and related phenotypes

Author Response

Dear Reviewer,

Sincere thanks should be given to the reviewer for the constructive comments and suggestions. The responses to the comments are given below. To be clearer and in accordance with the reviewer concerns, we have added the responses point-by-point as follows:

Major comments:

“Authors describe a pedigree of a family with autosomal dominant retinitis pigmentosa (ADRP) related to mutation in PRPF31. The article appear scientifically sound although reading is quite difficult due to poor English grammar. I suggest a strong revision of the text from a native English speaker because the topic is interesting and worth of consideration: it shows the potential of WGS, although we think that array-CGH would have been a viable option to show the described deletion.”

Response:

We are grateful for your suggestions. We will cherish the opportunity to re-write the manuscript. We are regretful for difficulty of understanding caused by the linguistic issues. This manuscript has been revised extensively according to your specific and constructive suggestions. In addition, the expression of the manuscript has been improved with the help of a native English speaker. We made our efforts to improve the language for better understanding the manuscript. We agree that array-CGH is helpful to detect the large deletion/duplication. It is an available analysis for situation like the large deletion.

Minor specific comments:

Comment 1:

  1. Introduction, page 2, line 46: 8000 genes associated to RP is questionable, up to now on RetNet RP is associated to about 300…

Response 1:

We are very sorry for our negligence of the wrong number. We were going to write 300 genes here but we made a mistake. The sentence was re-phrased. (Line 62)

Comment 2:

Introduction, page 2, line 65: … “may not present” seems more suitable than “…may not appear….”

Response 2:

Thanks for your advice. The phrase has been changed according to the comment. (Line 88)

Comment 3:

Results Section, page 5, lines 192-194: please remove the sample paragraph

Response 3

We are sorry for our negligence and have already fixed that.

Comment 4:

Results section, 3.1 paragraph, page 5, line 101: “…typical RP signs..” is maybe better than “…typical RP symptoms..”

Response 4

Thanks for your suggestion. We have modified the sentence to “…typical RP characteristics”. (Line 244)

Comment 5:

Results section, 3.1 paragraph, page 5, lines 204-206: “We found bull’s-eye maculopathy in fundus autofluorescence of macular and atrophy of retinal pigment epithelial cells and loss of the peripheral outer nuclear layer of retina in macular OCT in the youngest patient”

Please rephrase “We found bull’s-eye maculopathy in fundus autofluorescence, atrophy of retinal pigment epithelium and loss of the peripheral outer nuclear layer of retina in OCT scan of the youngest patient”: if it is corresponding to the phenotype described it sounds better than the description provided.

Response 5

Thanks a lot for your suggestion, which is highly appreciated. The sentence has been re-phrased according to the comment and merged with another sentence. (Line 250-252)

The description of OCT was stated in the Figure legend.

Comment 6:

Same paragraph, line 207: please correct “…have appeared..” with “manifested” or “presented”

Response 6

We have changed “appeared” to “presented”. (Line 251)

Comment 7:

For readers it would be useful an explicative image of the deletion to better see which genes contains and the distribution: moreover, there are many CNVs database (i.e. UCSC, decipher) but no information on their use is provided to compare similar deletions and related phenotypes.”

Response 7:

Thanks for your great suggestion on improving the readability of our manuscript. We have re-edited the Figure 3 in the Results section.

We sincerely hope that this revised manuscript has addressed all your comments and suggestions. We appreciated for reviewers’ warm work earnestly and hope that the correction will meet with approval. Once again, thank you very much for your comments and suggestions.

Sincerely,

The Authors

Reviewer 5 Report

Dear authors,

Congratulations for pursuing genetic understanding of such complex case in a large family. 

Nonetheless, extensive English editing is needed and this makes full comprehension of your work difficult. I will point out a few examples, but there are others:
- Line 28: "A special phenotype, early onset in all patients, has been firstly reported in RP11 family". Do you mean to say this is the first report or early onset retinal dystrophy in patients with mutations in PRPF31? Please make it clearer.
- Line 52: Not all geneticists and ophthalmologists are working on restoring vision in RP patients. Please rephrase.
- Line 198: I suggest saying that all patients presented the Early Onset Retinal Dystrophy phenotype. This a well known phenotype, not a special one. It may be unusual for this gene, but you cannot say it is a special phenotype in general. 
- Line 201: Symptom is something only the patient is able to perceive. Instead, fundoscopic findings would be called signs of RP.
- Line 207: Which report? Simply state the IOP by Tonopen.
- Line 244: Do not say "Normal people"! I suggest "healthy controls" instead.

Other corrections not related to English:

- Gene names must always be in italic.
- Line 46: Please check if this information of 8000 genes related to RP is correct. Have I understood the sentence correctly? Or is it 8000 variants in genes?
- Line 56: Please cite source that PRPF31 is the second most common gene causing AD RP.
- Line 58: You say "other genes involved in the splicing process" as if you have already mentioned this before, but this is not the case. Please correct. This paragraph is generally very confusing.
- Line 100: I suggest deleting the first sentence of this paragraph. 
- Line 132: You mention different numbers of patients than before, was this copied from somewhere else or have I not understood the methodology?
- Line 157: Another different number of patients is presented (14???). Was DNA not extracted from all your affected and unaffected family members?
- Line 191: Delete paragraph with instructions for writing the article. 
- Figure 2: Describe in figure legend what findings can be seen in the images. 
- Table 1: Remove genotype from the table title. No genotype is shown in this table. 
- I believe some methodology is discussed in the results. Simply state your results. Explanations should be in methodology or discussion.
- Line 309: Can the fact that there is a large deletion explain such a severe phenotype?

I suggest careful proofreading and extensive English editing. 

Author Response

Dear Reviewer,

Thank you very much for your time involved in reviewing the manuscript and your very encouraging comments on the manuscript. We have provided a point-by-point response to the reviewers' comments below in red color.

Comments on language and responses:

“Nonetheless, extensive English editing is needed and this makes full comprehension of your work difficult.”

We are grateful for your suggestions. We will cherish the opportunity to re-edit the manuscript. We are regretful for difficulty of understanding caused by the linguistic issues. This manuscript has been revised extensively according to your specific and constructive suggestions. In addition, the manuscript has been improved with the help of a native English speaker with extensive English editing. 

Comment 1:

- Line 28: "A special phenotype, early onset in all patients, has been firstly reported in RP11 family". Do you mean to say this is the first report on early onset retinal dystrophy in patients with mutations in PRPF31? Please make it clearer.

Response 1:

Thanks a lot for your comment. We have revised the text to address your concerns and hope that it is now clearer. (Line 35-37)

Comment 2:

- Line 52: Not all geneticists and ophthalmologists are working on restoring vision in RP patients. Please rephrase.

Response 2:

We are sorry for our wrong statement and have already fixed that. (Line 71-72)

Comment 3:

- Line 198: I suggest saying that all patients presented the Early Onset Retinal Dystrophy phenotype. This a well-known phenotype, not a special one. It may be unusual for this gene, but you cannot say it is a special phenotype in general. 

Response 3:

We fully agree with your comment and have re-phrased the sentence. (Line 240-242)

Comment 4:

- Line 201: Symptom is something only the patient is able to perceive. Instead, fundoscopic findings would be called signs of RP.

Response 4:

Sorry for our negligence and we have re-phrased it. (Line 243-244)

Comment 5:

- Line 207: Which report? Simply state the IOP by Tonopen.

Response 5:

The sentence was revised for better readability. (Line 254-255)

Comment 6:

 Line 244: Do not say "Normal people"! I suggest "healthy controls" instead.”

Response 6:

Sorry for the wrong phrase. We think the sentence is not suitable for the Results section. It was removed without loss of comprehension. (Line 301-302)

Other corrections not related to English and responses:

In the remainder of this letter, we discuss each of your comments on the other problems not related to English individually along with our corresponding responses.

Comment 1:

-Gene names must always be in italic.

Response 1:

We are very sorry for our incorrect writings. Thanks a lot for the comment. We have fixed the errors.

Comment 2:

- Line 46: Please check if this information of 8000 genes related to RP is correct. Have I understood the sentence correctly? Or is it 8000 variants in genes?

Response 2:

We are grateful for your kind remind. We were going to write 300 genes here but we made a mistake. The sentence was re-phrased. (Line 62)

Comment 3:

- Line 56: Please cite source that PRPF31 is the second most common gene causing ADRP.

Response 3:

Thank you for the suggestion. In fact, PRPF31 is the second most common gene in most population in some research. We re-wrote the statement and added the reference in the revised manuscript. (Line 75-76)

Comment 4:

- Line 58: You say "other genes involved in the splicing process" as if you have already mentioned this before, but this is not the case. Please correct. This paragraph is generally very confusing.

Response 4:

Thank you for the constructive comment. This paragraph has been revised. (Line 75-83)

Comment 5:

- Line 100: I suggest deleting the first sentence of this paragraph. 

Response 5:

We agree and have updated the paragraph. (Line 133-134)

Comment 6:

- Line 132: You mention different numbers of patients than before, was this copied from somewhere else or have I not understood the methodology?

Response 6:

Sorry for our mistake of writing the wrong number of patients. The DNA was extracted from all the family members and used for the mutation analysis. We have fixed the mistake. (Line 168-169)

Comment 7:

- Line 157: Another different number of patients is presented (14???). Was DNA not extracted from all your affected and unaffected family members?

Response 7:

We apologize for the wrong statement. We were going to state that the parametric linkage analysis was performed in the affected and unaffected family members. The software used for haplotype analysis was set on a complete penetrance mode in order to make sure the accurate pathogenic region. Because of the incomplete penetrance caused by PRPF31, the haplotype showed a strong linkage of pathogenic region and phenotype in 14 family members. The non-penetrant carriers were not included in the Figure 5. The paragraph has been revised. (2.6 Parametric linkage analysis and haplotype construction in Page 5)

Comment 8:

- Line 191: Delete paragraph with instructions for writing the article. 

Response 8

We are sorry for our negligence and have already removed it.

Comment 9:

- Figure 2: Describe in figure legend what findings can be seen in the images. 

Response 9

Thanks a lot for the constructive suggestion. We have added the description to the figure legend.

Comment 10:

- Table 1: Remove genotype from the table title. No genotype is shown in this table. 

Response 10

Sorry for our mistake. The “genotype” was removed from the table title. (Table 1)

Comment 11:

I believe some methodology is discussed in the results. Simply state your results. Explanations should be in methodology or discussion.

Response 11

To address the reviewer' s concern, we have revised the Results section.

Comment 12:

- Line 309: Can the fact that there is a large deletion explain such a severe phenotype?”

Response 12:

Yes, we agree with your comment. It may be the partial reason for severe phenotype including early onset of disease. We will further study this large deletion in model organs using iPS cells or animals. Thanks for your suggestive hypothesis.

We tried our best to improve the manuscript and made some changes in the manuscript. These changes will not influence the content and framework of the paper. And we did not list all the changes but marked in red in revised paper. We hope that the correction will meet with approval.

Once again, thank you very much for your comments and suggestions.

Sincerely,

The Authors

Round 2

Reviewer 1 Report

A study by Yuanzheng Lanet al. with a new current title “A 69-kb deletion in chr19q13.42 including PRPF31 gene in a 2 Chinese family affected with autosomal dominant retinitis pigmentosa” presents a large family of 5 generations with a history of retinal degeneration.  The authors identified a novel mutation in pedigree, causative of PRPF31-related adRP and proposed the mechanism of action by thorough ophthalmic phenotyping and genotyping using a wide range of genetic methods such as WGS, MLPA, linkage analysis and haplotype, RNA-seq.

The authors addressed all required comments, especially improving significantly structure and language. However, there are still some improvement to address, especially concerning the Discussion Section.

The remaining comments are, as follows:

1.     Row 446 in the revised version: correct to “ocular” in this sentence.

2.     Methods section: you need to clearly state why the ERGs were not possible to perform. ERG is a very important diagnostic method and gold standard for diagnosis of RP. Please explain why it was not performed in some of the individuals of the studied family.

3.     Figure 2, legend: make few edits in the text as follows: row 637- correct to “Fundus images presented bone spicules like pigmentation…”; row 638-639 – this statement is incomplete. Row 639: “small ring” not the “small circle” and please correct the entire statement, e. g. small ring with increase autofluorescence surrounding fovea and area with decreased autofluorescence….”

4.     Discussion, row 782: correct to “…have confirmed genetic diagnosis…”

5.     Discussion, sentence which starts with “Though the youngest patient (-3)…”- correct this statement, this is too long, e.g. divide it into two.

6.     Discussion, row 907: correct to “…detected through…”

7.     Discussion, row 928: correct to “the most likely candidate gene…”

8.     Discussion, row 929, please correct the statement starting with “Additionally, it shows typical incomplete penetrance phenotype??….”

9.     Discussion, row 1194: correct to “model”.

10.  Discussion, row 1392, please correct this sentence to “We sought to elucidate…”

11.  Discussion, row 1403-1405: please correct linguistic issue in this statement.

12.  Discussion Section is improved however still too lengthy and somehow unstructured. This applies specifically to paragraphs row 1191-1381. The connecting phrases are missing in between of the paragraphs, the text still does not flow well.

Author Response

Dear Reviewer:

We are glad to hear from you. We sincerely thank you for thoroughly examining our manuscript and providing very helpful and encouraging comments to guide our revision. We used “Track Changes” function of MS word to highlight our improvements. We also hope that our explanation can fully address all of your concerns. In the remainder of this letter, we discuss each of your comments individually along with our corresponding responses.

Responses to Comments:

Comment 1:

Row 446 in the revised version: correct to “ocular” in this sentence.

Response 1:

Sorry for our wrong spelling. It has been fixed.

Comment 2:

Methods section: you need to clearly state why the ERGs were not possible to perform. ERG is a very important diagnostic method and gold standard for diagnosis of RP. Please explain why it was not performed in some of the individuals of the studied family.

Response 2:

We agree that ERG is indeed the gold standard for RP diagnosis. Considering that all family members presented similar and typical characteristics of RP, ERGs were performed in some patients. Most family members reside in a remote village and live in poverty. Due to the extensive cost of time and money, some elderly patients presenting low vision/blindness refused to go to the hospital for extra examination. We got to the village and tried our best to perform the clinical examinations in the family with portable equipment. With the clinical examinations (including age of onset, manifestations, fundus changes, and OCT), we are convinced of the RP diagnosis of all affected family members.

Comment 3:

Figure 2, legend: make few edits in the text as follows: row 637- correct to “Fundus images presented bone spicules like pigmentation…”; row 638-639 – this statement is incomplete. Row 639: “small ring” not the “small circle” and please correct the entire statement, e. g. small ring with increase autofluorescence surrounding fovea and area with decreased autofluorescence….”

Response 3:

Thank you for the helpful comment! The figure legend has been revised.

Comment 4:

Discussion, row 782: correct to “…have confirmed genetic diagnosis…”

Response 4:

We agreed and made improvement according to your comment. It does make the sentence clearer and more readable.

Comment 5:

Discussion, sentence which starts with “Though the youngest patient (Ⅴ-3)…”- correct this statement, this is too long, e.g. divide it into two.

Response 5:

Thank you for the comment. It was changed to “Though the youngest patient (Ⅴ-3) with night blindness didn’t show a significant change of pigment, loss of peripheral PRC outer segment and RPE atrophy were identified in his macular OCT. These findings provide strong evidence for his RP diagnosis. ”

Comment 6:

Discussion, row 907: correct to “…detected through…”

Response 6:

We are sorry for the wrong writing and have corrected it. Thank you for your thoroughly examining and detailed comment.

Comment 7:

Discussion, row 928: correct to “the most likely candidate gene…”

Response 7:

Thank you for your suggestion. It has been re-written according to your comment.

Comment 8:

Discussion, row 929, please correct the statement starting with “Additionally, it shows typical incomplete penetrance phenotype??….”

Response 8:

Sorry for the wrong grammar. It was changed to “Additionally, members with mutated PRPF31 showed typical incomplete penetrance phenotype, which is supported by the haploinsufficiency theory.”

Comment 9:

 Discussion, row 1194: correct to “model”.

Response 9:

Sorry for our negligence. It has been fixed.

Comment 10:

Discussion, row 1392, please correct this sentence to “We sought to elucidate…”

Response 10:

Thanks a lot for the comment. It was re-vised according to the comment.

Comment 11:

Discussion, row 1403-1405: please correct linguistic issue in this statement.

Response 11:

The sentence has been changed to “The functions of genes in linkage region are related to immunology, transmembrane channel, or protein involved in regulation of gene expression.”

Comment 12:

Discussion Section is improved however still too lengthy and somehow unstructured. This applies specifically to paragraphs row 1191-1381. The connecting phrases are missing in between of the paragraphs, the text still does not flow well.

Response 12:

Thank you for the constructive comment. We have re-vised the Discussion section and added connecting phrases to improve the structure and make it more readable.

We would like to take this opportunity to thank you for all your time involved and this great opportunity for us to improve the manuscript. We hope you will find this revised version satisfactory.

Yours sincerely,

The Authors

Reviewer 5 Report

Thank you for putting an effort into the English editing. It has made your report much morr comotrhensible and interesting to read! 
Congrats!

Author Response

Dear Reviewer,

Glad to hear from you! We sincerely thank you for thoroughly examining our manuscript and providing very helpful and encouraging comments to guide our revision. We are glad that our explanation can address your concerns. We would like to thank you once again for this great opportunity for us to improve the manuscript.

Sincerely,

The Authors